# The Prevalence of Mild, Moderate, and Severe Nomophobia Symptoms: A Systematic Review, Meta-Analysis, and Meta-Regression

**DOI:** 10.3390/bs13010035

**Published:** 2022-12-30

**Authors:** Haitham Jahrami, Khaled Trabelsi, Omar Boukhris, Jumana Hasan Hussain, Ahmad F. Alenezi, Ali Humood, Zahra Saif, Seithikurippu R. Pandi-Perumal, Mary V. Seeman

**Affiliations:** 1Ministry of Health, Manama 410, Bahrain; 2Department of Psychiatry, College of Medicine and Medical Sciences, Arabian Gulf University, Manama 323, Bahrain; 3High Institute of Sport and Physical Education of Sfax, University of Sfax, Sfax 3000, Tunisia; 4Research Laboratory—Education, Motricity, Sport and Health, EM2S, LR19JS01, University of Sfax, Sfax 3000, Tunisia; 5SIESTA Research Group, School of Allied Health, Human Services and Sport, La Trobe University, Melbourne 3086, Australia; 6Ministry of Health, Sulaibkhat, Jamal Abdel Nasser Street, Kuwait 13001, Kuwait; 7Somnogen Canada Inc., College Street, Toronto, ON M5S 1A8, Canada; 8Saveetha Medical College and Hospitals, Saveetha Institute of Medical and Technical Sciences, Saveetha University, Chennai 602105, Tamil Nadu, India; 9Department of Psychiatry, University of Toronto, Toronto, ON M5S 1A8, Canada

**Keywords:** addiction, anxiety, fear of missing out, FOMO, iDisorder, nomophobia

## Abstract

NOMOPHOBIA, or NO MObile PHone Phobia, refers to a psychological condition in which people fear being disconnected from their mobile phones. The purpose of this review was to establish the prevalence of nomophobia symptoms in youth and young adults according to severity, country, culture, population, measurement tool, and year of data collection. An electronic search of fourteen databases, two digital preservation services, and three content aggregator services was conducted from the inception of each database until 15 September 2021. A total of 52 studies involving 47,399 participants from 20 countries were included in the analyses. The prevalence of nomophobia was defined as the proportion of individuals scoring at or above established cut-offs on validated measures. Based on a random-effects meta-analysis, approximately 20% of individuals showed mild symptoms of nomophobia, 50% showed moderate symptoms, and 20% showed severe symptoms. Our results showed that university students from non-Western cultures are the most likely to suffer severe symptoms. In the year 2021, the prevalence rate of nomophobia increased. The instrument that was best able to detect nomophobia was the nomophobia questionnaire. Most individuals who own mobile phones experience mild or moderate symptoms of nomophobia. Severe symptoms deserve attention from clinicians and research scientists. A valid method of identifying individuals with a severe addiction to their mobile phones will help with timely and effective therapeutic management.

## 1. Introduction

‘Nomophobia’ stands for “no-mobile-phone phobia”, an acronym first coined by the authors of a UK Postal Office study in 2008 [1]. The study found that, out of a sample of about 2000 adults who owned mobile phones, >50% experienced symptoms of anxiety when unable to access their phones [1]. These are the characteristic symptoms of this condition [2,3] and they appear regardless of the reasons for access failure (losing or misplacing one’s phone, loss of battery life, or finding oneself in an area with no network coverage) [2]. In spite of its name, this condition does not appear to be a phobia, but rather a behavioral addiction [2].

Research into the determinants of nomophobia is ongoing [3] but the most common unanswered research question, thus far, remains accurate prevalence, i.e., the establishment of specific numbers of individuals with the condition in any given population over a specific time period [4,5]. 

The Nomophobia Questionnaire (NMP-Q), developed by researchers at Iowa State University, is the most commonly used tool in prevalence studies [4,5]. The NMP-Q uses a Likert-like scale and, therefore, can quantify the severity of nomophobia [4,5]. Four main dimensions and/or causes are involved in nomophobia: (1) fear or nervousness associated with not being able to communicate with others; (2) fear of not connecting with others; (3) fear of not having immediate access to information; and (4) fear of giving up the comfort provided by mobile devices [4,5]. 

Nomophobia is connected with feelings of loneliness, low self-esteem, and unhappiness, particularly among young people [2,3,4]. The development of a significant reliance on mobile technology that produces continual diversions also impacts other elements of life. School, work, and general productivity are negatively influenced [1,4]. Furthermore, such reliance contributes to interpersonal distance and isolation, impacting relationships and interactions [3,5]. 

There are several reasons why a systematic review and meta-analysis of the prevalence rate of nomophobia is needed [6]. The first is to increase the power and precision of point prevalence estimates to understand the magnitude of the problem at any point in time [6]. The second reason is to identify the prevalence differences between subpopulations, for instance, between university students and age-matched non-students, to determine the distribution of preventive and interventive resources [6]. Perhaps the most important scientific reason is to attempt to resolve conflicting results among past studies [6]. For example, estimates for nomophobia symptoms (of various severity) have ranged between 25% [7] and 100% [8]. This may depend on the measuring tool, the targeted demographic, or cultural differences in the importance placed on mobile phone communication. It may also depend on public health policies relative to behavioral addictions.

Our team recognized the need for prevalence accuracy and conducted a systematic review and meta-analysis at the end of 2020 [2]. We analyzed 20 papers, involving about 12,500 participants from ten countries. The prevalence of mild, moderate, and severe nomophobia was about 25%, 50%, and 20%, respectively [2]. 

Many questions were left unanswered in the previous work. Is this condition becoming more prevalent? Does it occur with undue frequency in specific local communities? Does its incidence correlate with a suspected cause? Are there variations among countries, cultures (Western vs. Non-Western), demographics, measurement tools, and time periods? Can COVID-19 be held responsible for fluctuations? This possibility arises because several public health measures implemented by different governments in response to COVID-19 have increased the use of communication technology [9]. 

For these reasons, we decided to conduct an updated review using the PICO framework (Population, Intervention, Comparator, Outcomes) [10]; Population: individuals over the age of 12, Intervention: none, Comparators: (a) different countries, (b) Western vs. Non-Western cultures, (c) age and sex comparisons, (d) differences according to measurement tools, and (e) time period, Outcomes: determination of the prevalence rate of the three levels of nomophobia symptoms. 

## 2. Method

Before registering our protocol, we did a detailed analysis of PROSPERO and other evidence networks to prevent duplication. The protocol was then registered in the PROSPERO International Prospective Register of Systematic Reviews (PROSPERO) database (Registration number: CRD42022355657). Our protocol followed the PRISMA2020 protocol for systematic reviews and meta-analyses [11]. 

Two members of our team performed all independent electronic searches for pertinent studies published between the origin of each database and 15 September 2022. The following databases were searched: AccessMedicine, BIOSIS Citation Index, CINAHL, ClinicalKey, Cochrane Library (via Ovid), EMBASE, Health and Wellness (GALE), PROQUEST Research Library (including ABI/INFORM), Psychiatry Online, PsycINFO, PubMed (including MEDLINE), ScienceDirect, Scopus, and the Web of Science. There were no restrictions on language use. The search syntax, search methodology, and thorough search translations are displayed in Appendix A. 

To identify important grey literature [12], two digital preservation services, CLOCKSS and Swiss National Library (Helveticat), and three content aggregator services, Google Scholar, Scilit, and WorldCat (OCLC), were screened for best match hits. 

Based on PICO, key terms and PubMed Medical Subjects Headings (MeSH) were used as search terms. The (All Fields) search was constructed using the Boolean logic operators (OR, AND, NOT). The search was conducted using the following keywords: “nomophobia*”, (OR) “no-mo-phobia, (OR “ “no mobile* phobia, (OR) “ “mobile* phobia,” (OR) “mobile* addiction,” (AND) “prevalence”. The reference lists of the identified studies were examined to ensure that all pertinent publications had been covered. The final search results were transformed into a Microsoft Office (Excel spreadsheet 365 *.xlsx) file to filter and remove duplicates. The citations employing (Research Information Systems *.RIS) or integrated files were managed using EndNote 20.4.1.

Inclusion criteria were (1) original English-language papers about nomophobia published prior to 15 September 2022, (2) participants over the age of 12 (3) all participants completed a nomophobia screening test, and (4) participant responses to each test were scored and reported so that the percentages of participants falling above and below predetermined cut-off points could be calculated.

Exclusion criteria were (1) research targeting something other than the prevalence of nomophobia and (2) studies for which, despite contacting the authors, we were unable to obtain the information we needed The PRISMA 2020 study selection flowchart is depicted in Figure 1. 

### 2.1. Screening, Data Extraction, Quality Assessment, and Data Analysis

The studies chosen for the systematic review were screened and coded using ASReview [13], a free online tool that integrates digital technologies (such as natural language processing) with artificial intelligence and machine learning. The accuracy of abstract screening was improved using the semi-automated Abstrackr [14], an abstract screening tool for systematic reviews. Whenever required, data were extracted from plot images using the free and open-source web application WebPlotDigitizer v4.5 [15]. Quality check was maintained by manual cross-checking of the integrity of the data by a member of the extraction team. 

To standardize data extraction, three members of the study team (OB, AFA, and AH). independently extracted the following variables in addition to the primary finding of the event rate of nomophobia (by severity). Data extraction explicitly included the following information: author names, publication year, country of data collection, sample size, mean age (years), sex (male: female ratio), and the test used to assess if nomophobia was present or not. 

Disagreements as to what should be included/excluded were resolved through consensus among the aforementioned three reviewers. If an agreement could not be reached, a fourth author (ZS or HJ) was brought in to resolve the matter through discussion. If important information was lacking from a publication, the author of the article was contacted. 

A pair of authors (any of OB, AFA, or AH), working independently, used the Newcastle-Ottawa Scale (NOS) to assess the caliber of the included studies [16]. We used the NOS checklist created for cross-sectional investigations [17]. There are several components to it: participant selection, comparability, outcome, and statistics. Each item in the NOS is given one to three-quarters of a star on a rating scale [16]. As a result, cross-sectional studies can only receive a maximum score of nine. A study that receives an eight is considered of good quality with a low risk of bias, a study that receives a five to seven is considered of moderate quality with a low risk of bias, and a study that receives a zero to four is considered of poor quality with a high risk of bias. 

Using the random-effects model, a classic frequentist meta-analysis was conducted, assuming that actual effects will differ over time between samples [17]. The DerSimonian–Laird method was used to estimate and adjust for the variance of the effect between studies using the untransformed proportions and the general inverse variance method, with a continuity correction of 0.5 in studies with zero cell frequencies [18]. In random-effects modeling, the assumption is that different sets of studies estimate different, yet conceptually related, effects by using different measures. In each study, the pooled prevalence is reported along with the 95% confidence interval. Meta-analysis data were visualized using a forest plot [19]. Statistical analyses were conducted and presented according to the Meta-analysis of Observational Studies in Epidemiology (MOOSE) protocol [20]. 

An I^2^ value between 75–100% indicates a high degree of heterogeneity between studies [21]. A Cochran’s Q statistic [22] was also used to evaluate heterogeneity, as well as tau2 (*τ*^2^) and tau (*τ*) [21]. The Jackson method was used for the confidence interval of tau2 and tau. The H statistic [23] is equal to Cochran’s χ^2^ heterogeneity statistic divided by the degree of freedom. The initial visual tool used to investigate publication bias was a funnel plot [24]. An inverse Galbraith radial [25] plot was used to visualize heterogeneity by plotting observed effect sizes against their corresponding standard errors (horizontal axis). An arc shows the effect size or outcome on the right-hand side of a full-scale Galbraith plot [26,27]. The Doi plot [28] substitutes a folded normal quantile (Z-score) vs. effect plot for the traditional scatter (the funnel) plot [29] of precision versus effect. Studies make up the limbs of this plot; if there is asymmetry, one or more studies may make up one limb more than the other [30], causing an unequal divergence of both limbs from the midpoint [31]. In the absence of asymmetry, it would be anticipated that the Doi plot would be divided into two zones with comparable areas by a line drawn perpendicular to the *X*-axis from its tip. The gold standard for detecting publication bias was also used by employing rank correlations by Begg and Mazumdar [32] and Egger’s regression [33]. 

When outliers are included in meta-analyses, their validity and robustness may be compromised [34]. Studies classified as outliers, when their confidence interval did not match the pooled effects, were addressed by sensitivity analysis [34]. To ensure no inordinate influence was coming from a single study, we used the Jackknife sensitivity analysis. In this analysis, the main meta-analysis is repeated as many times as there are studies included, removing one study at a time [34].

The odds of a research paper being published are affected when its results encounter publication bias. An adjusted point estimate was generated using the trim and fill approach to correct funnel plot asymmetry due to publication bias [35]. 

For investigating heterogeneous outcomes and answering specific queries regarding distinct populations or study characteristics, subgroup meta-analyses [36] and meta-regression models [37] were used. Categorical variables, including country and culture (Western vs. non-Western), population (general adults vs. university students vs. adolescents), and measures/scales, were used in subgroup analyses. United Nations regional groups of member states were used to categorize Western and non-Western countries [38]. Study subgroups were based on the year of publication (year of data collection) to investigate the effect of time as a confounder. Forest plots were then used to present results for each subgroup meta-analysis. 

A meta-regression involves predicting the outcome variable based on one or more explanatory factors [37]. The regression coefficient of a meta-regression will show how the outcome variable changes as the explanatory variable (maybe a moderator or effect modifier or confounding variable) increase by one unit [37]. Meta-regression was performed on univariate analysis using age and sex. A sex-age interaction term was tested. In statistically significant meta-regression models, effect sizes were reported using R^2^. A small effect size was defined as 1–8%, a medium effect size as 9–24%, and a large effect size was defined as 25% [39].

All data were analyzed using R software version 4.1.3 for statistical computing [40]. A *p*-value < 0.05 was considered statistically significant. To perform the classical meta-analysis, the packages ‘meta’ [41] and ‘metafor’ [42] were used. Using the package ‘forester’, a summary forest plot was generated based on the combined effect sizes of multiple forest plots, omitting the results of individual studies. For example, the combined results (omitting results from individual studies) from mild, moderate, and severe nomophobia were all presented in one plot instead of three separate forest plots.

For quality assessment, risk-of-bias plots were generated using the package ‘robvis’ [43]. A summary plot (weighted) shows the proportion of information inside each judgment. A detailed risk of bias assessment of all studies, displayed using a traffic light plot, depicts the bias risk in each domain and the overall risk.

### 2.2. Role of the Funding Source

This systematic review and meta-analysis have received no funding from the government, private sector, or non-profit sector.

## 3. Results

### 3.1. The Characteristics of the Included Studies

The search included the time frame from the inception of the databases until 15 September 2022. A total of 791 records were located using various sources, including electronic database searches. After duplicate records were eliminated, 459 records remained. The title, abstract and full content of all potential articles were examined. A PRISMA2020 flowchart is used to represent the search process in Figure 1. 

A total of 52 studies [7,8,44,45,46,47,48,49,50,51,52,53,54,55,56,57,58,59,60,61,62,63,64,65,66,67,68,69,70,71,72,73,74,75,76,77,78,79,80,81,82,83,84,85,86,87,88,89,90,91,92,93] (53 data points) involving 47,399 participants from 20 countries were involved in this meta-analysis. The number of studies (Ks) and corresponding overall sample size per country (Ns) are as follows in alphabetical order: Australia (K = 3, *N* = 6601), Bahrain (K = 3, *N* = 1752), Bosnia and Herzegovina (K = 1, *N* = 1083), Canada (K = 3, *N* = 2481), China (K = 1, *N* = 473), Croatia (K = 1, *N* = 257), Ghana (K = 1, *N* = 345), India (K = 5, *N* = 2262), Iran (K = 1, *N* = 320), Italy (K = 4, *N* = 5719), Kuwait (K = 1, *N* = 512), Lebanon (K = 1, *N* = 2260), Oman (K = 1, *N* = 740), Pakistan (K = 4, *N* = 940), Peru (K = 1, *N* = 3139), the Philippines (K = 1, *N* = 3374), Saudi Arabia (K = 3, *N* = 6314), Spain (K = 1, *N* = 850), Thailand (K = 1, *N* = 638), and Turkey (K = 16, *N* = 7339). A summary of the studies included in this systematic review and meta-analysis is in Table 1. 

It was interesting to observe that although the term nomophobia was first described in 2008, the first measurement tool (i.e., NMP-Q) was published in 2015–2016, and all prevalence studies were published in the year 2018 or afterward. 

The mean prevalence sample size was 895 (95%CI 613; 1175) participants. Males accounted for 37% (95%CI 33; 42%), and the mean age was 22 (95%CI 21; 25) years. A total of 48 (91%) of the studies used the NMP-Q and mostly adopted a cross-sectional approach for data collection. Online surveying was used in 50 (95%) of the studies. The studies were generally robust with a mean quality score of 7.3 (95% 7.0; 7.6). A detailed quality assessment of each of the included studies is available in a traffic light plot format in Appendix A. The risk of bias was moderate in 16 (30%) of the studies and the remaining 37 (70%) were of a low risk of bias. Most of the risk of bias was in the sample selection. Detailed results are in a summary plot format in Figure 2. 

Out of the 53 data points, 53 (100%) provided a global estimate of all nomophobia symptoms and 42 (80%) provided an estimate of nomophobia by severity (mild, moderate, and severe forms of the condition). 

### 3.2. Prevalence of Nomophobia by Severity

Figure 3 provides a summary of the entire results of this meta-analysis on the prevalence of nomophobia by severity, country, culture, population, measurement tool, and year of data. The following section provides a detailed examination of each element at a micro level. 

#### 3.2.1. All Symptoms (Cumulative or All Severities)

A random-effects meta-analysis evaluated the prevalence of nomophobia in all populations (K = 53, *N* = 47,399) and generated a pooled prevalence rate of 93.92% (93.19; 94.66%), 95%PI (88.56; 99.29%), *τ*^2^ = 0.007 (0.0038; 0.0101); *τ* = 0.0265 (0.0614; 0.1006), I^2^ = 99.6% (99.5%; 99.6%); H = 15.38 (14.82; 15.95), Q = 12,293.72 (df = 52) *p* < 0.001. Detailed results are presented in Table 2. The Forest plot of all nomophobia symptoms is in Appendix A. 

A (leave-one-out) sensitivity analysis found that no study had a greater than 2% impact on the global prevalence estimate. Some outliers were detected but deleting them did not result in a major change in the estimates (within 3%) of all nomophobia symptoms. 

Visual inspection of the funnel plot (Appendix A), Galbraith plot (Appendix A), and DOI plot (Appendix A) indicated a publication bias. A linear regression test of funnel plot asymmetry showed a test result: *t* = −4.55, df = 51, *p*-value < 0.001, suggesting publication bias. Similarly, the rank correlation test of funnel plot asymmetry showed a test result of *z* = −7.00, *p*-value < 0.001, suggesting the presence of publication bias. The adjusted meta-analysis for all nomophobia symptoms (using the trim and fill approach) yielded an estimate of 99.78% (98.86%; 100.00%). Detailed results are presented in Table 3, Part 1. 

For all nomophobia symptoms, a statistically significant difference was observed based on country, culture, population, measurement tool, and year of data collection, all *p* < 0.05. Detailed results are presented in Table 3, Part 1. 

Bahrain and Canada had the highest rates of all nomophobia symptoms with prevalence rates of 100.00% (99.86%; 100.00%) and 100.00% (99.91%; 100.00%), respectively. India and Saudi Arabia had the lowest rates of 85.74% (80.36%; 91.11%) and 83.49% (64.23%; 100.00%), respectively. 

Western cultures had a higher prevalence rate of all nomophobia symptoms with a rate of 95.30% (94.04%; 96.56%) vs. Non-Western cultures of 93.38% (92.44%; 94.32%), *p* = 0.02. Detailed results are presented in Table 3, Part 1.

University students appeared to have the highest prevalence of all nomophobia symptoms with a rate of 97.38% (96.72%; 98.04%), followed by the general adult population at 95.15% (93.06%; 97.25%), and high school students and community adolescents 84.17% (82.11%; 86.22%). The difference between the three population groups was statistically significant, *p* < 0.001. Detailed results are presented in Table 3, Part 1.

The NMP-Q captured a larger prevalence rate compared to other tools, with estimates of 97.59% (97.13%; 98.05%) vs. 66.52% (53.66%; 79.37%), respectively. The difference between the two groups was statistically significant, *p* < 0.001. Detailed results are presented in Table 3, Part 1.

Analysis of all nomophobia symptoms by year showed an uptrend between 2018–2021 (peak 2021), followed by a slight downtrend trend to date. Prevalence rates for 2018, 2019, 2020, 2021, and 2022 were: 82.30% (76.20%; 88.40%), 85.35% (81.71%; 89.00%), 94.77% (92.13%; 97.42%), 99.46% (99.12%; 99.79%), and 93.01% (91.55%; 94.47%). The difference in all nomophobia symptoms between the years was statistically significant, *p* < 0.001. Detailed results are presented in Table 3, Part 1.

Subgroup meta-analyses for all nomophobia symptoms by country, culture, population measurement tool, and year of data collection are shown in Appendix A. 

Meta-regression models showed that both age (in years) and sex (proportion of male participants) are statistically significant predictors for all nomophobia symptoms. Age showed: *p* < 0.001; *τ*^2^ (estimated amount of residual heterogeneity): 0.007 (SE = 0.003); *τ* (square root of estimated *τ*^2^ value): 0.0270; I^2^ (residual heterogeneity/unaccounted variability): 99.59%; H^2^ (unaccounted variability/sampling variability): 240.99, and R^2^ (amount of heterogeneity accounted for): 0.00%. Sex showed: *p* < 0.001; *τ*^2^ (estimated amount of residual heterogeneity): 0.007 (SE = 0.003); *τ* (square root of estimated *τ*^2^ value): 0.0272; I^2^ (residual heterogeneity/unaccounted variability): 99.58%; H^2^ (unaccounted variability/sampling variability): 240.74; and R^2^ (amount of heterogeneity accounted for): 0.00%. The age-sex interaction term was not a significant predictor *p* = 0.6095. Detailed results are presented in Table 3, Part 1. 

#### 3.2.2. Mild Symptoms

Mild nomophobia was prevalent in 25.80% (19.83; 31.78%), 95%PI (00.00; 65.99%), *τ*^2^ = 0.0386 (0.0182; 0.0776); *τ* = 0.1965 (0.1348; 0.2785), I^2^ = 99.8%; H = 20.81 (20.13; 21.51), Q = 17,747.07 (df 41) *p* < 0.001. Detailed results are presented in Table 2. A Forest plot of mild nomophobia symptoms is in Appendix A. 

A (leave-one-out) sensitivity analysis found that no study had a greater than 2% impact on the global prevalence estimate. Some outliers were detected but deleting them did not result in a major change in the estimates (within 3%) of mild nomophobia symptoms. 

Visual inspection of the funnel plot, Galbraith plot, and DOI plot indicated a publication bias. A linear regression test of funnel plot asymmetry showed a test result: *t* = 7.05, df = 40, *p*-value < 0.001, suggesting publication bias. However, the rank correlation test of funnel plot asymmetry showed a test result of *z* = −1.22, *p*-value = 0.2206 showing the absence of publication bias. The adjusted meta-analysis for mild nomophobia symptoms (using a trim and fill approach) yielded an estimate of 4.17% (1.00%; 9.97%). Detailed results are presented in Table 3, Part 2. 

For mild nomophobia symptoms, a statistically significant difference was observed based on the country only, *p* < 0.001. The difference in mild nomophobia symptoms based on culture, population, measurement tool, and year of data collection did not reach a statistical significance of *p* > 0.05. Detailed results are presented in Table 3, Part 2. Subgroup meta-analyses for mild nomophobia symptoms by country, culture, population measurement tool, and year of data collection are shown in Appendix A. 

Meta-regression models showed that both age (in years) and sex (proportion of male participants) were not significant predictors for mild nomophobia symptoms. Age showed: *p* = 0.4722; *τ*^2^ (estimated amount of residual heterogeneity): 0.0396 (SE = 0.0297); *τ* (square root of estimated *τ*^2^ value): 0.1989; I^2^ (residual heterogeneity/unaccounted variability): 99.77%; H^2^ (unaccounted variability/sampling variability): 443.55; R^2^ (amount of heterogeneity accounted for): 0.00%. Sex showed: *p* = 0.8009; *τ*^2^ (estimated amount of residual heterogeneity): 0.0342 (SE = 0.0197); *τ* (square root of estimated *τ*^2^ value): 0.1850; I^2^ (residual heterogeneity/unaccounted variability): 99.65%; H^2^ (unaccounted variability/sampling variability): 287.95; R^2^ (amount of heterogeneity accounted for): 11.34%. The age-sex interaction term was not a significant predictor, *p* = 0.9404. Detailed results are presented in Table 3, Part 2. 

#### 3.2.3. Moderate Symptoms

Moderate nomophobia was prevalent in 52.40% (44.21; 60.60%), 95%PI (00.00; 100.00%), *τ*^2^ = 0.0728 (0.0288; 0.0787); *τ* = 0.2698 (0.1697; 0.2806), I^2^ = 99.7%; H = 17.86 (17.21; 18.53), Q = 13,080.70 (df = 41) *p* < 0.001. Detailed results are presented in Table 2. The Forest plot of all nomophobia symptoms is presented in Appendix A. 

A (leave-one-out) sensitivity analysis found that no study had a greater than 2% impact on the global prevalence estimate. Some outliers were detected but deleting them did not result in a major change in the estimates (within 4%) of moderate nomophobia symptoms. 

Visual inspection of the funnel plot, Galbraith plot, and DOI plot indicated a publication bias. A linear regression test of funnel plot asymmetry showed a test result of *t* = −2.74, df = 40, and *p*-value = 0.0092, suggesting publication bias. However, the rank correlation test of funnel plot asymmetry showed a test result of *z* = 1.54, *p*-value = 0.1238, showing the absence of publication bias. The adjusted meta-analysis for moderate nomophobia symptoms (using a trim and fill approach) yielded an estimate of 74.66 % (65.59%; 83.73%). 

For moderate nomophobia symptoms, a statistically significant difference was observed based on the country and the measurement tool used, *p* < 0.001. The difference in moderate nomophobia symptoms based on culture, population, and year of data collection did not reach a statistical significance of *p* > 0.05. Detailed results are presented in Table 3, Part 3. Subgroup meta-analyses for all nomophobia symptoms by country, culture, population measurement tool, and year of data collection are shown in Appendix A. 

Meta-regression models showed that both age (in years) and sex (proportion of male participants) were not significant predictors for mild nomophobia symptoms. Age showed: *p* = 0.8940; *τ*^2^ (estimated amount of residual heterogeneity): 0.0745 (SE = 0.0294); *τ* (square root of estimated *τ*^2^ value): 0.2730; I^2^ (residual heterogeneity/unaccounted variability): 99.69%; H^2^ (unaccounted variability/sampling variability): 323.37; R^2^ (amount of heterogeneity accounted for): 0.00%. Sex showed: *p* = 0.8198; *τ*^2^ (estimated amount of residual heterogeneity): 0.0700 (SE = 0.0261); *τ* (square root of estimated *τ*^2^ value): 0.2646; I^2^ (residual heterogeneity/unaccounted variability): 99.66%; H^2^ (unaccounted variability/sampling variability): 292.81; R^2^ (amount of heterogeneity accounted for): 3.86%. The age-sex interaction term was not a significant predictor, *p* = 0.7915. Detailed results are presented in Table 3, Part 3.

#### 3.2.4. Severe Symptoms

Severe nomophobia was prevalent in 20.35% (16.51; 24.20%), 95%PI (00.00; 46.07%), *τ*^2^ = 0158 (0.0124; 0.0468); *τ* = 0.1257 (0.1116; 0.2163), I^2^ = 99.6%; H = 15.75 (15.13; 16.40), Q = 10,176.69 (df = 41) *p* < 0.001. Detailed results are presented in Table 2. The Forest plot of all nomophobia symptoms is presented in Appendix A. 

A (leave-one-out) sensitivity analysis found that no study had a greater than 1% impact on the global prevalence estimate. Some outliers were detected but deleting them did not result in a major change in the estimates (within 2%) of severe nomophobia symptoms. 

Visual inspection of the funnel plot, Galbraith plot, and DOI plot indicated a publication bias. Linear regression test of funnel plot asymmetry showed a test result of *t* = 7.35, df = 40, and *p*-value < 0.001, suggesting publication bias. However, the rank correlation test of funnel plot asymmetry showed a test result of *z* = −0.98, *p*-value = 0.3293, showing the absence of publication bias. The adjusted meta-analysis for severe nomophobia symptoms (using a trim and fill approach) yielded an estimate of 4.62 % (0.77%; 8.47%). Detailed results are presented in Table 3, Part 4. 

For severe nomophobia symptoms, a statistically significant difference was observed based on country, culture, measurement tool, and year of data collection with *p* < 0.001. The difference in moderate nomophobia symptoms based on population did not reach a statistical significance of *p* > 0.05. Detailed results are presented in Table 3, Part 4. Subgroup meta-analyses for all nomophobia symptoms by country, culture, population measurement tool, and year of data collection are shown in Appendix A. 

Meta-regression models showed that both age (in years) and sex (proportion of male participants) were not significant predictors for mild nomophobia symptoms. Age showed: *p* = 0.3732; *τ*^2^ (estimated amount of residual heterogeneity): 0.0187 (SE = 0.0124); *τ* (square root of estimated *τ*^2^ value): 0.1366; I^2^ (residual heterogeneity/unaccounted variability): 99.61%; H^2^ (unaccounted variability/sampling variability): 253.91; R^2^ (amount of heterogeneity accounted for): 0.00%. 

Sex showed: *p* = 0.9139; *τ*^2^ (estimated amount of residual heterogeneity): 0.0203 (SE = 0.0104); *τ* (square root of estimated *τ*^2^ value): 0.1426; I^2^ (residual heterogeneity/unaccounted variability): 99.56%; H^2^ (unaccounted variability/sampling variability): 227.25; R^2^ (amount of heterogeneity accounted for): 0.001%. The age-sex interaction term was not a significant predictor, *p* = 0.6594. Detailed results are presented in Table 3, Part 4.

## 4. Discussion

Prevalence rates are core indicators of the healthcare needs of a population. These rates are also essential inputs for the burden of disease studies and simulation models that make projections about population health in the future. It is for this reason that population nomophobia rates are needed. There may be a question as to why severity rates are required in clinical settings. It is important to know the proportion of severe nomophobia because, in all probability, only severe cases will need clinical intervention. 

The main findings of this systematic review and meta-analysis show that nomophobia symptoms are very common in all studied populations. Most individuals experience symptoms that are mild or moderate; mild, moderate, and severe symptoms are present in about 25%, 50%, and 20% of the individuals. Our results suggest that university students, from non-Western countries and cultures, are the ones most affected by severe symptoms. A wave of increased prevalence of nomophobia was observed in the year 2021, potentially related to COVID-19. The NMP-Q appeared to be the most sensitive tool for the detection of moderate and severe nomophobia. 

The findings of the present systematic review and meta-analysis on the prevalence rate of nomophobia are consistent with the results of two previous meta-analyses [2,94]. Thus, the results of mild, moderate, and severe symptoms of about 25%, 50%, and 20%, respectively, are very robust. 

Due to its serious impact on health, severe nomophobia needs to be the focus of the discussion, since it can lead to mental disorders [50], with symptoms of depression [95], anger [96], nervousness, anxiety, and stress [72,97,98], aggression [83], and insomnia and other sleep issues [70]. Recent studies also showed that excessive and intensive use of smartphones can lead to musculoskeletal problems [99,100]. The main problems identified have been neck [99] and thumb [100] injuries. Severe nomophobia is associated with personal safety issues, including an increased risk of road traffic accidents [74,101]. 

Special situations may trigger increased nomophobia symptoms. For example, a recent study looked at how a social media outage affects nomophobia [102]. Results revealed that symptoms of nomophobia, notable anxieties about access and connectivity, increased dramatically during outages [102]. Resultant insomnia also increases the severity of nomophobia [102]. 

Our results suggest that neither age nor sex is a significant predictor of nomophobia. According to a systematic review published in 2021, however, there has been a recent significant increase in nomophobia among women and those aged under 35 [94]. However, it is difficult to be certain because of methodological differences among studies [94]. The proportion of “at-risk” individuals and those suffering from nomophobia varies greatly from study to study, ranging from 13 to 79% and 6 to 73%, respectively [94]. Similar to our findings, the review previously mentioned supports the greater frequency of moderate cases relative to severe cases [94]. 

It has been reported that adolescents who experience sleep loss and ongoing sleep deprivation due to nomophobia may experience behavioral and academic issues (e.g., possible aggression, failing grades, and absenteeism) [70]. A greater likelihood of accidents has also been predicted. Concern has been expressed that the lifestyle and quality of life of teenagers will be severely compromised if nomophobia is left unattended by physicians and psychologists [2]. 

### Limitations and Directions for Future Research

This systematic review and meta-analysis has a limitation inherited from the original studies included in the analysis and related to the fact that most of the participants were limited to a narrow age range. To better estimate nomophobia’s distribution among different age groups, future studies should include older adults and the elderly. Nevertheless, as the world moves toward a more digital lifestyle, this systematic review provides an understanding of the global prevalence of this newly emerging condition, which may well increase with time. For this reason, efforts at prevention and timely intervention are in order. 

There are a few other issues with this meta-analysis. First, heterogeneity was high. In a large epidemiological meta-analysis, this is to be expected. The use of random-effects modeling was expected to address concerns linked to the consequences of reviewing a large number of studies that do not all follow the same pattern, but follow a distribution. To mitigate this, we used 95% prediction intervals. Individual patient data (IPD) meta-analyses are useful and should be promoted in future studies to work out, assess, and discuss different elements of nomophobia. Second, we only found a few moderators. Future evaluations should broaden this investigation to include additional lifestyle variables such as physical activity, smoking, and substance use, with a focus on controlling for pre-existing stress-related illnesses such as posttraumatic stress disorder, adjustment disorders, anxiety, and depression. Risk factors would be important to determine. 

Human beings are increasingly living a socially isolated digital lifestyle, which impacts behavior and undermines coping mechanisms. Preventive measures for this increasingly global issue are needed. Due to its rising prevalence, it is suggested that this disorder be included in forthcoming editions of the Diagnostic and Statistical Manual of Mental Disorders (DSM), the International Classification of Diseases (ICD), and the International Classification of Sleep Disorders (ICSD).

Future studies are encouraged to adopt a two-phase design for improved prevalence estimation. This would mean screening a population sample, then interviewing varying proportions of screened individuals that have been stratified according to their probability of fulfilling the criteria for nomophobia. Prevalence rates can then be calculated by weighting back to the original population. 

## 5. Conclusions

The prevalence rates of mild, moderate, and severe nomophobia, as here reviewed, are approximately 25%, 50%, and 20%, respectively, across the world regions represented in the available studies. University students appear to be the most impacted by the disorder. For any medical condition, and this includes nomophobia, knowing prevalence rates helps in the determination of risk factors, which then permits the development of a program of prevention, early intervention, and effective therapeutic management. 

## Figures and Tables

**Figure 1 behavsci-13-00035-f001:**
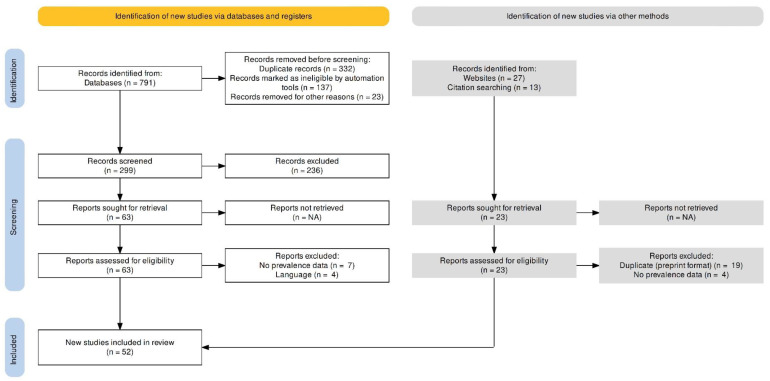
PRISMA 2020 flow diagram for study selection.

**Figure 2 behavsci-13-00035-f002:**
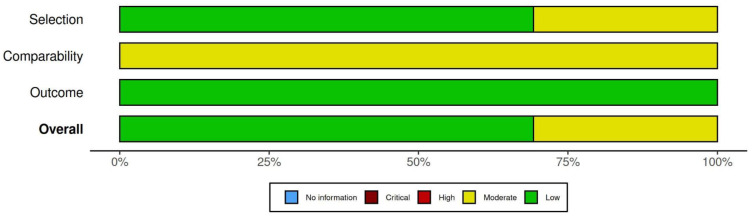
A summary plot of the risk of bias assessment.

**Figure 3 behavsci-13-00035-f003:**
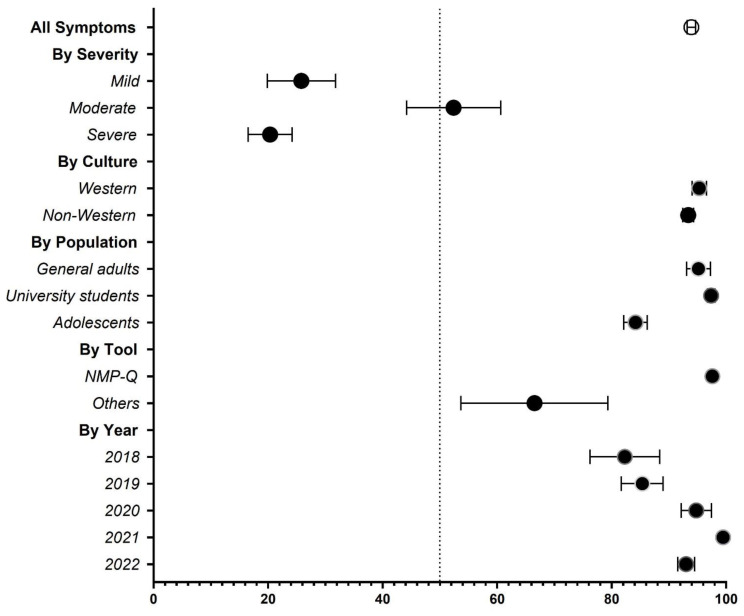
Summary forest plot of nomophobia.

**Table 1 behavsci-13-00035-t001:** Summary of studies included in the systematic review and meta-analysis about the prevalence of nomophobia symptoms.

SN	Ref	Study	Country	Population	Sample (*N*)	Male:Female	Age (Years)	Tool	Nomophobia (%)	NOS (Stars)
1	[44]	Al-Balhan, 2018	Kuwait	University Students	512	50:50%	20	NMP-Q	100.0	8
2	[45]	Almarzooqi, 2022	Saudi Arabia	General Population	893	74:26%	24	NMP-Q	99.4	8
3	[46]	Alwafi, 2022	Saudi Arabia	General Population	5191	31:69%	24	Others	51.0	7
4	[47]	Ayar, 2018	Turkey	University Students	755	17:83%	21	NMP-Q	99.7	8
5	[48]	Bano, 2021	Saudi Arabia	Adolescents	230	47:53%	22	NMP-Q	100.0	6
6	[49]	Bartwal, 2020	India	University Students	451	38:62%	21	NMP-Q	100.0	8
7	[50]	Bragazzi, 2019	Italy	University Students	403	40:60%	28	NMP-Q	100.0	8
8	[51]	Buctot, 2021	Philippines	Adolescents	3374	42:58%	15	NMP-Q	99.5	8
9	[52]	Catone, 2020	Italy	Adolescents	2959	52:48%	15	Others	69.0	7
10	[53]	Çelik İnce, 2021	Turkey	University Students	607	25:75%	21	NMP-Q	99.7	8
11	[54]	Çevik-Durmaz, 2021	Turkey	University Students	234	18:82%	22	NMP-Q	100.0	6
12	[55]	Çırak, 2022	Turkey	University Students	451	33:67%	20	NMP-Q	100.0	8
13	[56]	Copaja-Corzo, 2022	Peru	University Students	3139	39:61%	22	NMP-Q	96.0	8
14	[57]	Coskun, 2020	Turkey	General Population	210	51:49%	33	NMP-Q	98.1	6
15	[58]	Daei, 2019	Iran	University Students	320	41:59%	23	Others	100.0	5
16	[59]	Denprechavong, 2022	Thailand	University Students	638	82:18%	20	NMP-Q	76.2	8
17	[60]	Essel, 2022	Ghana	General Population	345	43:57%	20	NMP-Q	100.0	6
18	[61]	Farchakh, 2021	Lebanon	General Population	2260	0:100%	28	NMP-Q	97.7	8
19	[62]	Farooq, 2022	Pakistan	University Students	455	31:69%	22	NMP-Q	100.0	8
20	[63]	Farooqui, 2018	India	University Students	145	46:54%	19	NMP-Q	100.0	6
21	[64]	Fidanci, 2021	Turkey	University Students	386	51:49%	22	NMP-Q	96.6	8
22	[65]	Gurbuz, 2020	Turkey	General Population	400	42:58%	28	Others	100.0	7
23	[66]	Hoşgör, 2021	Turkey	General Population	178	10:90%	31	NMP-Q	96.1	6
24	[67]	Işcan, 2021	Turkey	University Students	641	27:73%	21	NMP-Q	99.7	8
25	[68]	Jahrami, 2021	Bahrain	General Population	549	46:54%	27	NMP-Q	100.0	8
26	[69]	Jahrami, 2021	Bahrain	General Population	654	46:54%	27	NMP-Q	100.0	8
27	[70]	Jahrami, 2022	Bahrain	General Population	549	49:51%	27	NMP-Q	100.0	8
28	[71]	Jilisha, 2019	India	University Students	774	41:59%	19	NMP-Q	98.8	8
29	[72]	Kaur, 2021	Pakistan	University Students	209	52:48%	21	NMP-Q	100.0	6
30	[73]	Kaviani, 2020	Australia	General Population	2838	47:53%	25	NMP-Q	99.2	8
31	[74]	Kaviani, 2022	Australia	General Population	2773	47:53%	20	NMP-Q	99.2	8
32	[75]	Koppel, 2022	Australia	General Population	990	30:70%	51	NMP-Q	98.9	8
33	[76]	Kundu, 2022	India	University Students	338	50:50%	21	NMP-Q	100.0	6
34	[77]	Lupo, 2020	Italy	General Population	540	27:73%	33	NMP-Q	91.3	8
35	[78]	Ma, 2021	China	University Students	473	32:68%	19	NMP-Q	82.9	8
36	[79]	Polat, 2022	Turkey	Adolescents	745	24:76%	21	NMP-Q	100.0	8
37	[7]	Prasad, 2017	India	University Students	554	47:53%	22	Others	24.9	7
38	[80]	Qutishat, 2020	Oman	University Students	740	34:66%	33	NMP-Q	99.3	8
39	[81]	Ramos-Soler, 2021	Spain	Adolescents	850	52:48%	15	NMP-Q	100.0	8
40	[82]	Santl, 2022	Croatia	Adolescents	257	14:86%	22	NMP-Q	100.0	6
41	[83]	Schwaiger, 2020	Pakistan	University Students	138	33:67%	20	NMP-Q	100.0	6
42	[84]	Schwaiger, 2022	Pakistan	University Students	138	33:67%	20	NMP-Q	97.1	6
43	[85]	Sevim-Cirak, 2021	Turkey	Adolescents	1066	32:68%	20	NMP-Q	100.0	8
44	[86]	Sui, 2022	Canada	University Students	258	20:80%	22	NMP-Q	100.0	6
45	[8]	Sui, 2022	Canada	University Students	1002	21:79%	23	NMP-Q	100.0	8
46	[87]	Sui, 2022	Canada	University Students	1221	28:72%	23	NMP-Q	100.0	8
47	[88]	Tomczyk, 2022	Bosnia and Herzegovina	Adolescents	1083	40:60%	15	NMP-Q	29.5	8
48	[89]	Torpil, 2021	Turkey	University Students	181	15:85%	20	NMP-Q	100.0	6
49	[90]	Torpil, 2022	Turkey	University Students	46	33:67%	21	NMP-Q	100.0	6
50	[91]	Torpil, 2022	Turkey	University Students	215	10:90%	23	NMP-Q	100.0	6
51	[92]	Yavuz, 2019	Italy	Adolescents	1817	46:54%	15	NMP-Q	99.2	8
52	[93]	Yildiz Durak, 2019	Turkey	Adolescents	612	52:48%	13	Others, NMP-Q	53.4	7

Notes: NMP-Q = Nomophobia Questionnaire; NOS = Newcastle-Ottawa Scale.

**Table 2 behavsci-13-00035-t002:** Results of the random-effects meta-analysis models of the prevalence of nomophobia symptoms.

Analysis	Descriptive		Random-Effects Meta-Analysis	Adjusted Meta-Analysis	Heterogeneity						Publication Bias		Moderators		
	K	*N*	Pooled Results (95%CI)		I^2^	*τ* ^2^	*τ*	H	Q	*p*	Egger’s Test	Rank Test	Age	Sex	Int
Prevalence of all nomophobia symptoms (all severity)	53	47,399	93.92% (93.19%; 94.66%)	99.78 % (98.86%; 100.00%)	99.6%	0.001	0.03	15.38	12,293.72 (df = 52)	0.001	0.001	0.001	0.001	0.001	0.61
Prevalence of all nomophobia symptoms (mild symptoms only)	42	33,780	25.80% (19.83%; 31.78%)	04.17 % (01.00%; 09.97%)	99.8%	0.04	0.20	20.81	17,747.07 (df = 41)	0.001	0.001	0.22	0.47	0.80	0.94
Prevalence of all nomophobia symptoms (moderate symptoms only)	42	33,780	52.40% (44.21%; 60.60%)	74.66 % (65.59%; 83.73%)	99.7%	0.07	0.27	17.86	13,080.70 (df = 41)	0.001	0.01	0.12	0.89	0.82	0.79
Prevalence of all nomophobia symptoms (severe symptoms only)	42	33,780	20.35% (16.51%; 24.20%)	04.62 % (00.77%; 08.47%)	99.6%	0.02	0.13	15.75	10,176.69 (df = 41)	0.001	0.001	0.33	0.37	0.91	0.66

Notes: K: Represents the number of included studies. *N*: Represents the number of included samples of the included studies. Rank-based nonparametric data augmentation was done using the Duval and Tweedie trim and fill approach. The technique was used to calculate the number of studies that a meta-analysis was missing since the most extreme results were suppressed on one side of the funnel plot. I^2^: Refers to the percentage of variation across samples due to heterogeneity rather than chance. *τ*^2^: Describes the extent of variation among the effects observed in different samples (between-sample variance). *τ*: Under the presumption that these genuine effect sizes are normally distributed, tau is an estimate of the standard deviation of the distribution of true effect sizes. The prediction interval is computed using tau. H: Describes confidence intervals of heterogeneity. It is more broadly characterized by the method of moments. As an inherited technique from meta-analysis, it is utilized in meta-regression.

**Table 3 behavsci-13-00035-t003:** Results of the random-effects subgroup meta-analysis models of the prevalence of nomophobia symptoms by country, culture, population, tool, and year.

Part 1—All Symptoms								
Analysis	Descriptive		Random-Effects Meta-Analysis	Heterogeneity				
	K	*N*	Pooled Results (95%CI)	I^2^	*τ* ^2^	*τ*	Q	*p*
**By Country**								
Australia	3	6601	99.18% (98.96%; 99.40%)	0.0%	0.001	0.001	0.86	0.001
Bahrain	3	1752	100.00% (99.86%; 100.00%)	0.0%	0.001	0.001	0.001	0.001
Canada	3	2481	100.00% (99.91%; 100.00%)	0.0%	0.001	0.001	0.001	0.001
India	5	2760	85.74% (80.36%; 91.11%)	99.8%	0.001	0.06	1667.12	0.001
Italy	4	5719	89.96% (83.49%; 96.42%)	99.8%	0.001	0.07	1317.43	0.001
Pakistan	4	940	99.94% (99.53%; 100.00%)	26.5%	0.001	0.001	4.08	0.001
Saudi Arabia	3	6314	83.49% (64.23%; 100.00%)	100.0%	0.03	0.17	4568.08	0.001
Turkey	16	7339	96.29% (95.35%; 97.24%)	98.7%	0.001	0.02	1195.41	0.001
**By Culture**								
Western	8	10,372	95.30% (94.04%; 96.56%)	99.5%	0.001	0.02	1501.00	0.02
Non-Western	45	37,027	93.38% (92.44%; 94.32%)	99.6%	0.001	0.03	10,771.87	0.02
**By Population**								
General adults	14	18,370	95.15% (93.06%; 97.25%)	99.7%	0.001	0.04	5048.44	0.001
University students	28	15,424	97.38% (96.72%; 98.04%)	98.7%	0.001	0.02	2121.92	0.001
High school students and community adolescents	11	13,605	84.17% (82.11%; 86.22%)	99.8%	0.001	0.03	5071.85	0.001
**By Tool**								
NMP-Q	47	37,975	97.59% (97.13%; 98.05%)	98.8%	0.001	0.02	3844.29	0.001
Others	6	9424	66.52% (53.66%; 79.37%)	99.9%	0.03	0.16	7961.51	0.001
**By Year**								
2018	4	1966	82.30% (76.20%; 88.40%)	99.8%	0.001	0.06	1661.68	0.001
2019	6	4538	85.35% (81.71%; 89.00%)	99.6%	0.001	0.04	1163.34	0.001
2020	8	8276	94.77% (92.13%; 97.42%)	99.5%	0.001	0.04	1352.49	0.001
2021	15	11,892	99.46% (99.12%; 99.79%)	92.4%	0.001	0.01	184.09	0.001
2022	20	20,727	93.01% (91.55%; 94.47%)	99.8%	0.001	0.03	7886.39	0.001
**Part 2—Mild Symptoms**								
**Analysis**	**Descriptive**		**Random-effects meta-analysis**	**Heterogeneity**				
	**K**	** *N* **	**Pooled results (95%CI)**	**I^2^**	** *τ* ^2^ **	** *τ* **	**Q**	** *p* **
**By Country**								
Australia	3	6601	36.24% (33.83%; 38.66%)	74.6%	0.001	0.02	7.88	0.001
Bahrain	3	1752	6.45% (5.30%; 7.60%)	0.0%	0.001	0.001	0.03	0.001
India	4	1708	19.13% (15.94%; 22.33%)	61.4%	0.001	0.03	7.77	0.001
Italy	3	2760	51.79% (36.42%; 67.16%)	98.1%	0.02	0.13	107.81	0.001
Pakistan	4	940	10.17% (8.24%; 12.10%)	0.0%	0.001	0.001	1.78	0.001
Saudi Arabia	2	1123	17.45% (15.23%; 19.67%)	0.0%	0.001	0.001	0.05	0.001
Turkey	13	5881	28.68% (18.03%; 39.33%)	99.3%	0.04	0.19	1809.78	0.001
**By Culture**								
Western	5	5934	46.70% (8.91%; 84.48%)	99.9%	0.19	0.43	7683.16	0.22
Non-Western	37	27,846	22.95% (18.24%; 27.66%)	99.2%	0.02	0.14	4341.57	0.22
**By Population**								
General adults	13	13,179	28.57% (19.58%; 37.55%)	99.4%	0.03	0.16	1988.47	0.82
University students	23	12,519	24.36% (14.11%; 34.60%)	99.7%	0.06	0.25	8590.29	0.82
High school students and community adolescents	6	8082	25.30% (11.04%; 39.55%)	99.7%	0.03	0.18	1854.42	0.82
**By Tool**								
NMP-Q	40	33,060	26.10% (19.95%; 32.25%)	99.8%	0.04	0.20	17,607.12	0.07
Others	2	720	19.86% (16.95%; 22.77%)	0.0%	0.001	0.001	0.01	0.07
**By Year**								
2018	3	1412	23.47% (11.72%; 35.23%)	96.0%	0.01	0.10	50.33	0.65
2019	4	3314	32.91% (19.43%; 46.38%)	98.5%	0.02	0.14	197.64	0.65
2020	7	5317	31.79% (18.36%; 45.22%)	99.1%	0.03	0.18	693.36	0.65
2021	14	11,658	24.20% (17.33%; 31.07%)	98.9%	0.02	0.13	1168.32	0.65
2022	14	12,079	22.86% (12.23%; 33.49%)	99.9%	0.04	0.20	8692.36	0.65
**Part 3—Moderate symptoms**								
**Analysis**	**Descriptive**		**Random-effects meta-analysis**	**Heterogeneity**				
	**K**	** *N* **	**Pooled results (95%CI)**	**I^2^**	** *τ* ^2^ **	** *τ* **	**Q**	** *p* **
**By Country**								
Australia	3	6601	48.69% (47.48%; 49.90%)	0.0%	0.001	0.001	0.08	0.001
Bahrain	3	1752	72.95% (70.87%; 75.03%)	0.0%	0.001	0.001	0.01	0.001
India	4	1708	60.78% (54.40%; 67.15%)	85.2%	0.001	0.06	20.26	0.001
Italy	3	2760	36.31% (18.72%; 53.90%)	98.8%	0.02	0.15	164.49	0.001
Pakistan	4	940	56.13% (49.52%; 62.75%)	74.2%	0.001	0.06	11.63	0.001
Saudi Arabia	2	1123	51.47% (48.55%; 54.39%)	0.0%	0.001	0.001	0.63	0.001
Turkey	13	5881	47.95% (25.86%; 70.04%)	99.8%	0.16	0.41	7057.22	0.001
**By Culture**								
Western	5	5934	40.44% (19.85%; 61.04%)	99.6%	0.05	0.23	1080.70	0.23
Non-Western	37	27,846	54.03% (45.66%; 62.39%)	99.6%	0.07	0.26	9897.15	0.23
**By Population**								
General adults	13	13,179	53.01% (45.73%; 60.30%)	98.6%	0.02	0.13	871.24	0.80
University students	23	12,519	50.48% (41.23%; 59.72%)	99.2%	0.05	0.22	2849.63	0.80
High school students and community adolescents	6	8082	58.49% (35.16%; 81.81%)	99.9%	0.08	0.29	3811.99	0.80
**By Tool**								
NMP-Q	40	33,060	51.40% (42.93%; 59.87%)	99.7%	0.07	0.27	13,028.81	0.001
Others	2	720	72.38% (69.12%; 75.65%)	0.0%	0.001	0.001	0.34	0.001
**By Year**								
2018	3	1412	55.11% (50.89%; 59.34%)	55.2%	0.001	0.03	4.46	0.95
2019	4	3314	54.06% (42.90%; 65.22%)	97.3%	0.01	0.11	112.58	0.95
2020	7	5317	52.88% (39.56%; 66.20%)	98.9%	0.03	0.18	530.86	0.95
2021	14	11,658	52.19% (45.97%; 58.42%)	97.8%	0.01	0.12	585.36	0.95
2022	14	12,079	51.20% (31.59%; 70.81%)	99.9%	0.14	0.37	10,854.13	0.95
**Part 4—Severe symptoms**								
**Analysis**	**Descriptive**		**Random-effects meta-analysis**	**Heterogeneity**				
	**K**	** *N* **	**Pooled results (95%CI)**	**I^2^**	** *τ* ** ^2^	** *τ* **	**Q**	** *p* **
**By Country**								
Australia	3	6601	14.08% (12.23%; 15.94%)	77.4%	0.001	0.01	8.84	0.001
Bahrain	3	1752	20.60% (18.71%; 22.50%)	0.0%	0.001	0.001	0.05	0.001
India	4	1708	19.67% (15.77%; 23.57%)	73.7%	0.001	0.03	11.42	0.001
Italy	3	2760	8.77% (5.11%; 12.42%)	89.8%	0.001	0.03	19.65	0.001
Pakistan	4	940	31.68% (23.14%; 40.22%)	86.7%	0.01	0.08	22.63	0.001
Saudi Arabia	2	1123	30.95% (27.31%; 34.59%)	29.0%	0.001	0.02	1.41	0.001
Turkey	13	5881	22.45% (14.12%; 30.78%)	99.8%	0.02	0.15	5522.01	0.001
**By Culture**								
Western	5	5934	10.22% (5.19%; 15.25%)	97.7%	0.001	0.06	172.29	0.001
Non-Western	37	27,846	21.75% (17.25%; 26.26%)	99.6%	0.02	0.14	9784.91	0.001
**By Population**								
General adults	13	13,179	16.66% (11.43%; 21.88%)	99.1%	0.01	0.09	1312.05	0.33
University students	23	12,519	23.65% (15.37%; 31.93%)	99.4%	0.04	0.20	3877.84	0.33
High school students and community adolescents	6	8082	15.81% (6.35%; 25.27%)	99.7%	0.01	0.12	1639.16	0.33
**By Tool**								
NMP-Q	40	33,060	20.99% (17.01%; 24.98%)	99.6%	0.02	0.13	10,153.29	0.001
Others	2	720	7.70% (5.75%; 9.65%)	0.0%	0.001	0.001	0.67	0.001
**By Year**								
2018	3	1412	20.34% (11.53%; 29.16%)	93.3%	0.01	0.07	29.63	0.04
2019	4	3314	12.41% (6.23%; 18.58%)	96.4%	0.001	0.06	82.29	0.04
2020	7	5317	12.78% (6.09%; 19.47%)	99.0%	0.01	0.09	575.38	0.04
2021	14	11,658	21.29% (17.05%; 25.54%)	97.2%	0.01	0.08	464.88	0.04
2022	14	12,079	25.22% (15.70%; 34.75%)	99.8%	0.03	0.18	6590.38	0.04

Notes: K: Represents the number of included studies. *N*: Represents the number of included samples of the included studies. Rank-based nonparametric data augmentation was done using the Duval and Tweedie trim and fill approach. The technique was used to calculate the number of studies that a meta-analysis was missing since the most extreme results were suppressed on one side of the funnel plot. I^2^: Refers to the percentage of variation across samples due to heterogeneity rather than chance. *τ*^2^: Describes the extent of variation among the effects observed in different samples (between-sample variance). *τ*: Under the presumption that these genuine effect sizes are normally distributed, tau is an estimate of the standard deviation of the distribution of true effect sizes. The prediction interval is computed using tau. H: Describes confidence intervals of heterogeneity. It is more broadly characterized by the method of moments. As an inherited technique from meta-analysis, it is utilized in meta-regression.

## Data Availability

Derived data (and analysis codes) supporting the findings of this review are available from the corresponding author based on request.

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
