# Peer review of "The Prevalence of Mild, Moderate, and Severe Nomophobia Symptoms: A Systematic Review, Meta-Analysis, and Meta-Regression"

_behavsci, 2022, doi:10.3390/bs13010035_

Round 1

Reviewer 1 Report

This manuscript is systematic review, meta-analysis, and meta-regression of the nomophobia  symptoms' prevalence. The main idea behind the article is very good. However, there are certain areas that require significant improvements in order to be published.

Title of the manuscript is a bit of a reach. Only 20 countries were included and some parts of the world were not represented at all. Therefore, I would omit mentioning „world“ in the title. 

Introduction and Discussion need more work. In this format they are quite short, and they do not reflect study's foundation nor results in the best way. Please expand your introduction and in discussion reflect bit more on your results and interaction with current literature. 

Even though figures are mentioned throughout the text, I was no table to find them in the system. I believe they were not submitted. Also, supplemental material 10 is repeating in the Legend. It is very hard to go through the supplemental materials due to the lack of corresponding descriptions in the file. 

In the Method section, please change headings “2.4. Procedure”.

Visual presentation of the data in the tables is bit chaotic. While I do understand the need for extensive length due to the number of revised literature, I still believe that data could be presented in more understandable and clean way. 

Overall, this is promising work. However, there are quite a lot of pieces missing and if they are added appropriately, I believe it could be considered for publication. In this current form, that is not the case.

Author Response

This manuscript is systematic review, meta-analysis, and meta-regression of the nomophobia symptoms' prevalence. The main idea behind the article is very good. However, there are certain areas that require significant improvements in order to be published.

We thank you and the reviewers for taking the time to review our manuscript and provide constructive feedback that helped improve the presentation of our paper and demonstrate its originality.

Title of the manuscript is a bit of a reach. Only 20 countries were included and some parts of the world were not represented at all. Therefore, I would omit mentioning „world“ in the title. 

Authors’ response: We have omitted the word "world" in the title. The title now reads as: "The prevalence of mild, moderate, and severe nomophobia symptoms: a systematic review, meta-analysis, and meta-regression."

Introduction and Discussion need more work. In this format they are quite short, and they do not reflect study's foundation nor results in the best way. Please expand your introduction and in discussion reflect bit more on your results and interaction with current literature. 

Authors’ response: We have expanded the introduction by about 25% reaching now about 650 words we also expanded the discussion by about 25% reaching now about 1000 words. The full manuscript text is now about 6000 words.

In the introduction we added the following paragraphs:

Four main dimensions and/or causes are involved in nomophobia: (1) fear or nervousness associated with not being able to communicate with others; (2) fear of not connecting with specific others; (3) fear of not having immediate access to information; and (4) fear of giving up the comfort  provided by the mobile device[4,5].

Nomophobia is connected with feelings of loneliness, low self-esteem, and unhappiness, particularly among young people [2-4]. By developing a significant reliance on mobile technology and producing continual diversions, this negatively impacts other elements of life, such as work, school,  and general productivity [1,4]. Furthermore, it contributes to interpersonal distance and isolation, impacting relationships and interactions [3,5].

In the discussion we added the following paragraphs by bringing information about latest studies pertaining to nomophobia:

“Due to its serious impact on health, severe nomophobia needs to be the focus of the discussion, since it can lead to mental disorders [50], with symptoms not only  of nervousness, anxiety, and stress [72, 97, 98] but also depression [95], anger [96], aggression [83], insomnia and other sleep issues [70]. Recent studies also showed that, as a consequence of excessive and intensive use of smartphones, musculoskeletal problems can arise [99,100]. The main problems identified have been neck [99] and thumb [100] injuries. Severe nomophobia is associated with adverse effects on personal safety including an increased risk for road traffic accidents while driving [74,101].

Special situations may trigger increased nomophobia symptoms. For example, a recent study looked at how a social media outage affects nomophobia [102]. Results revealed that symptoms of nomophobia, notably anxieties about access and connectivity, increased dramatically during the outage [102]. Consequent insomnia increased nomophobia [102].

We also expanded on limitations offering suggestions for future work:

“This systematic review and meta-analysis has a limitation inherited from the original studies included in the analysis and related to the fact that most of the participants were limited to a narrow  age range.. To better estimate nomophobia's distribution among different age groups, future studies should include older adults and the elderly. Nevertheless, as the world moves toward a more digital lifestyle, this systematic review provides an understanding of the global prevalence of this modern emerging condition, which may well increase with time. For this reason, efforts at prevention and timely intervention are in order.

There are a few other issues with this meta-analysis. First, heterogeneity was high.  . In a large epidemiological meta-analysis, this is to be expected. The use of random-effects modeling was expected to address concerns linked to the consequences of reviewing a large number of studies that do not all follow the same pattern, but rather follow a distribution. To mitigate this, we used 95% prediction intervals. Individual patient data (IPD) meta-analyses are useful and should be promoted in future studies to work out, assess, and discuss different elements of nomophobia. Second, we only had a few moderators. Future evaluations should broaden this investigation to include additional lifestyle variables such as physical activity, smoking, and substance use, with a focus on controlling for stress-related illnesses such as posttraumatic stress disorder, adjustment disorders, anxiety, and depression. Risk factors are important to determine.

Even though figures are mentioned throughout the text, I was not able to find them in the system. I believe they were not submitted. Also, supplemental material 10 is repeating in the Legend. It is very hard to go through the supplemental materials due to the lack of corresponding descriptions in the file. 

Authors’ response: We have added Figure 1, Figure 2, Figure 3 at the end of the manuscript for peer-review purposed. During production we will work collaboratively with the mdpi production team to ensure the final production is more appealing as per mdpi policies.

We ensured that supplemental material 10 is presented only once in the Legend.

The supplemental materials will be presented in the final paper user hyperlink and therefore will be in form of click to show. We added list of the Legend to the zipped file to facilitate peer-review process.

In the Method section, please change headings “2.4. Procedure”.

Authors’ response: The headings “2.4. Procedure” changed to “2.4. Screening, data extraction, quality assessment, and data analysis”.

Visual presentation of the data in the tables is bit chaotic. While I do understand the need for extensive length due to the number of revised literatures, I still believe that data could be presented in more understandable and clean way. 

Authors’ response: We have switched the orientation of the visual presentation of the data in the tables from portrait to landscape. The tables are not presented in a more understandable and clean way. We will work collaboratively with the mdpi production team to ensure the final production is more appealing as per mdpi policies.

Overall, this is promising work. However, there are quite a lot of pieces missing and if they are added appropriately, I believe it could be considered for publication. In this current form, that is not the case.

Authors’ response: We thank you for your compliment. We have taken all steps to improve presentation and ensure that all parts are clearly presented as per the best standards.

Reviewer 2 Report

See attached

Author Response

The prevalence of mild, moderate, and severe nomophobia symptoms across the world: a systematic review, meta-analysis, and meta-regression. The manuscript reports on an interesting and lesser examined topic; the prevalence of nomophobia. Overall, the manuscript is well-written and with some minor revisions would be suitable for publication.

We thank you and the reviewers for taking the time to review our manuscript and provide constructive feedback that helped improve the presentation of our paper and demonstrate its originality.

Some suggestions:

Abstract

o Could the authors mention how many final identified studies (and how many countries) the prevalence data came from.

Authors’ response: we added the following information “A total of 52 studies  involving 47,399 participants from 20 countries were included in the analyses.

 Introduction

o Page 2 paragraph 1 correct the trigger for anxiety symptoms suggests, instead, a behavioral addiction.

Authors’ response: we corrected this from “Although the name of this condition implies a phobia, it is really a behavioral addiction [2/

o Page 2 paragraph 2 - The Nomophobia Questionnaire (NMP-Q) developed by Iowa State University re-searchers to identify nomophobia is the most commonly used research tool used in prevalence studies [4,5]. Would read better as: The Nomophobia Questionnaire (NMP-Q) developed by researchers at Iowa State University is the most commonly used tool in prevalence studies [4,5].

Authors’ response: we changed the sentence as suggested: “The Nomophobia Questionnaire (NMP-Q) developed by researchers at Iowa State University is the most commonly used tool in prevalence studies [4,5].”

o Page 2 paragraph 3 remove for instance

Authors’ response: for instance was removed from the paragraph.

o Page 2 paragraph 4 & 5 consider merging these paragraphs. When authors say Many questions were not answered in pervious work Do they mean in their own previous work? Then better to merge these paragraphs and say this earlier work.

Authors’ response: we corrected this to “Many questions were left unanswered in previous work”.

 Method

o Methods is mostly thorough.

Authors’ response: Thank you for your nice compliment. Once again, we thank you and the reviewers for taking the time to review our manuscript and provide constructive feedback that helped improve the presentation of our paper and demonstrate its originality.

o Page 3 paragraph 4 For consistency, could this paragraph start with Inclusion criteria were        

Authors’ response: Page 3 paragraph 4 For consistency, now start with “Inclusion criteria were:”

o Page 3 paragraph 7 insert a full stop after (by severity). Should the final sentence I this paragraph say and the assessment tool for severity of nomophobia rather than the test used to assess if nomophobia was present or not.

Authors’ response: we inserted a full stop after (by severity).

o I have not commented on the meta-analytic techniques used as I do not feel qualified to provide feedback. It reads well to me.

Authors’ response: Thank you for sincere comment. We publish high quality systematic reviews and meta-analysis heavily we assure the reviewer that mathematical/statistical techniques used are robust.

o Page 5 Section 2.7 Role of the funding source Should this be at the end of the paper?

Authors’ response: We added this to the end of the paper. But according to PRISMA20202 this is also need to be in the main paper.

 Results

o Page 5 Section 3.1 Reads A total of 791 records in total were located    Remove repetition.

Authors’ response: we changed “A total of 791 records in total were located” into “A total of 791 records was located”.

o Page 6 paragraph 4 Is there a typo Visual inspection of to funnel plot    Should it be the? See also Page 7 paragraph 4.

Authors’ response: Thank you for this correction we removed of  it now reads as “Visual inspection of the funnel”.

o Need to italicise all t, p, z

Authors’ response: all t, p, z are italicized.

o Page 6 Paragraph 9 Reads The NMP-Q captured a larger prevalence rate compared to other tools           Should the other tools be names here? As a minimum should they be named in the Notes section for Table 3 i.e., Others = xxxxxxx  

Authors’ response: Notes section for Table 3 was moved to the end and no reptations is available now.

Discussion

o Page 9 paragraph 2 Reads One might ask: why are severity rates clinically required? This chatty tone detracts from the scholarly nature of the paper. Suggest authors reword.

Authors’ response: We changed “One might ask: why are severity rates clinically required?” Into formal phrasing “There may be a question as to why severity rates are required in clinical settings.”

o Page 9 paragraph 5 check formatting i.e., irregular line spacing and font size.

Authors’ response: Font was set for Palatino Linotype 10 point for all manuscript.

o Page 9 paragraph 8 Check grammar This systematic review and meta-analysis have a limitation should be, has not have. Perhaps use a Subheading to introduce Limitations and Directions for Future Research.

Authors’ response:  Thank you for your comments. We added a complete section about - Limitations and Directions for Future Research – as shown below:

“Limitations and Directions for Future Research

“This systematic review and meta-analysis has a limitation inherited from the original studies included in the analysis and related to the fact that most of the participants were limited to a narrow  age range.. To better estimate nomophobia's distribution among different age groups, future studies should include older adults and the elderly. Nevertheless, as the world moves toward a more digital lifestyle, this systematic review provides an understanding of the global prevalence of this modern emerging condition, which may well increase with time. For this reason, efforts at prevention and timely intervention are in order.

There are a few other issues with this meta-analysis. First, heterogeneity was high.  . In a large epidemiological meta-analysis, this is to be expected. The use of random-effects modeling was expected to address concerns linked to the consequences of reviewing a large number of studies that do not all follow the same pattern, but rather follow a distribution. To mitigate this, we used 95% prediction intervals. Individual patient data (IPD) meta-analyses are useful and should be promoted in future studies to work out, assess, and discuss different elements of nomophobia. Second, we only had a few moderators. Future evaluations should broaden this investigation to include additional lifestyle variables such as physical activity, smoking, and substance use, with a focus on controlling for stress-related illnesses such as posttraumatic stress disorder, adjustment disorders, anxiety, and depression. Risk factors are important to determine.

 Tables

o Table 2 Try converting this table to landscape as some columns are not readable.

Authors’ response: All tables changed to landscape orientation.

o Table Notes are unnecessarily tabbed across and would look better if left aligned under each table.

Authors’ response: Notes are now left aligned under each table.

Thank you for allowing me to review this work. The authors have done an excellent job detailing their procedure and have provided new knowledge in a growing area nomophobia. I encourage the authors to revise their manuscript and wish them well with publishing their work.

Authors’ response: Thank you for your nice compliment. Once again, we thank you and the reviewers for taking the time to review our manuscript and provide constructive feedback that helped improve the presentation of our paper and demonstrate its originality.

Round 2

Reviewer 1 Report

Thank you for addressing my comments. I believe that revised version of your manuscript is worthy of publication.